# Spindle assembly checkpoint-dependent mitotic delay is required for cell division in absence of centrosomes

KC Farrell[1†], Jennifer T Wang[1‡], Tim Stearns[1,2*§]

[1]Department of Biology, Stanford University, Stanford, United States; [2]Department of Genetics, Stanford University School of Medicine, Stanford, United States

*For correspondence:
tstearns@rockefeller.edu

Present address: [†]Institute for Genetics and Cologne Excellence Cluster on Cellular Stress Responses in Aging-Associated Diseases (CECAD), University of Cologne, Cologne, Germany; [‡]Department of Biology, Washington University in St. Louis, St. Louis, United States; [§]The Rockefeller University, New York, United States

Competing interest: The authors declare that no competing interests exist.

**Abstract** The spindle assembly checkpoint (SAC) temporally regulates mitosis by preventing progression from metaphase to anaphase until all chromosomes are correctly attached to the mitotic spindle. Centrosomes refine the spatial organization of the mitotic spindle at the spindle poles. However, centrosome loss leads to elongated mitosis, suggesting that centrosomes also inform the temporal organization of mitosis in mammalian cells. Here, we find that the mitotic delay in acentrosomal cells is enforced by the SAC in a MPS1-dependent manner, and that a SAC-dependent mitotic delay is required for bipolar cell division to occur in acentrosomal cells. Although acentrosomal cells become polyploid, polyploidy is not sufficient to cause dependency on a SAC-mediated delay to complete cell division. Rather, the division failure in absence of MPS1 activity results from mitotic exit occurring before acentrosomal spindles can become bipolar. Furthermore, prevention of centrosome separation suffices to make cell division reliant on a SAC-dependent mitotic delay. Thus, centrosomes and their definition of two spindle poles early in mitosis provide a 'timely two-ness' that allows cell division to occur in absence of a SAC-dependent mitotic delay.

## eLife assessment

This work explores how centrosomes, which function as the primary microtubule organizing center in animal cells, regulate cell division by examining the process in cells in which centrosome formation has been inhibited. The carefully conducted experiments provide **convincing** support for the **important** observation that elongated, but successful, mitosis observed in cells lacking centrosomes is due to delays in cell cycle progression.

## Introduction

In eukaryotic cells, depolymerization of microtubules and the resultant mitotic arrest before anaphase led to the discovery of the 'mitotic block', (*Levan, 1938*; *Rieder and Palazzo, 1992*) now called the spindle assembly checkpoint (SAC). SAC activity prevents progression from metaphase to anaphase by inhibiting the anaphase-promoting complex/cyclosome (APC/C) (*Fang, 2002*; *Muhua et al., 1998*; *Nilsson et al., 2008*; *Fang et al., 1998*). The SAC, comprised of components largely enriched near kinetochores, (*Waters et al., 1998*; *Chen et al., 1996*; *Wang and Burke, 1995*; *Gurden et al., 2018*) becomes activated when defects in chromosome attachment are present (*Waters et al., 1998*; *Chen et al., 1996*; *O'Connell et al., 2008*; *Li and Benezra, 1996*). Thus, perturbation of spindle microtubules or certain microtubule motors leads to SAC activation (*Waters et al., 1998*; *Chen et al., 1996*; *Wang and Burke, 1995*; *Janssen et al., 2018*; *Kapoor et al., 2000*). At the other end of each side of the spindle apparatus, centrosomes are attached to and focus the spindle poles in a dynein-dependent manner (*Heald et al., 1997*; *Walczak et al., 1998*). Centrosomes are usually comprised

of two centrioles and pericentriolar material (PCM) that contains many microtubule nucleation factors that stabilize microtubule minus-ends, and they are major loci of microtubule minus ends in the mitotic spindles of mammalian cells (*Heald et al., 1997*; *Doxsey et al., 1994*; *Mitchison and Kirschner, 1984*; *Barr et al., 2010*; *Stearns and Kirschner, 1994*). However, in the absence of centrosomes, cultured mammalian cells can still divide bipolarly (*Khodjakov et al., 2000*; *Mahoney et al., 2006*; *Berns and Richardson, 1977*). Although centrosomes, when present, are major sites of microtubule nucleation, microtubules can be nucleated in mammalian cells by acentrosomal microtubule organizing centers (MTOCs; *Watanabe et al., 2020*; *Chinen et al., 2021*) as well as the chromatin-mediated microtubule nucleation pathway (*Heald et al., 1996*; *Caudron et al., 2005*; *Petry et al., 2011*). Nevertheless, absence of centrosomes alters bipolar spindle formation, such that spindles are monopolar early in mitosis and then resolve to become bipolar (*Watanabe et al., 2020*; *Chinen et al., 2021*; *Chinen et al., 2020*). The construction of such acentrosomal spindles occurs during a mitosis that is, on average, elongated in time (*Wong et al., 2015*; *Meitinger et al., 2016*). In this work, we tested whether SAC activity is necessary for this elongation, since the SAC is a well-conserved mechanism to delay mitotic progression. Furthermore, we asked whether the elongation is necessary for acentrosomal division to proceed and, if so, what centrosomes provide to negate the need for this elongation.

## Results and discussion
### MPS1 activity is required for mitotic elongation and cell division in acentrosomal cells

We removed centrosomes from human cells to interrogate their contributions to the temporal progression of mitosis. Acentriolar, hTERT-immortalized RPE-1 cells were generated though 10 d treatment with the PLK4 inhibitor centrinone (125 nM; *Wong et al., 2015*) or through deletion of *SASS6* which encodes a structural protein (SASS6) that templates centriole duplication (*Wang et al., 2015*). Acentriolar RPE-1 cells were generated in *TP53*[-/-] background, since RPE-1 cells with an intact p53 pathway undergo G1 arrest in absence of centrioles (*Wong et al., 2015*; *Meitinger et al., 2016*). Acentriolar wild-type U2OS cells, which are already p16-deficient (*Koh et al., 1995*) were generated through 10 d treatment with centrinone (125 nM). RPE-1 *TP53*[-/-] and U2OS cells treated with centrione showed significant increases in acentrosomal cells as compared to controls, as did *TP53*[-/-]; *SASS6*[-/-] cells in comparison to *TP53*[-/-] cells as measured by number of puncta positive for both γ-tubulin and centrin-3, which mark PCM and centrioles, respectively (*Figure 1a and a'*). RPE-1 *TP53*[-/-] and U2OS cells treated with centrinone and *TP53*[-/-]; *SASS6*[-/-] cells will be referred to as 'acentrosomal cells' for brevity. As has been previously reported (*Wong et al., 2015*; *Meitinger et al., 2016*), the time required to complete mitosis was elongated in acentrosomal cells compared to controls (*Figure 1b*). However, the mitoses were not uniformly elongated during each stage. Rather, the time from nuclear envelope breakdown (NEBD) through anaphase onset was elongated in acentrosomal cells, while the duration of anaphase through cytokinesis was not significantly different compared to controls (*Figure 1b and b'*).

The SAC is responsive to spindle defects that occur from NEBD through anaphase onset, (*Mansfeld et al., 2011*) the elongated mitotic period in acentrosomal cells. Therefore, we tested whether mitotic elongation in acentrosomal cells depended on SAC activity. We inhibited MPS1 (multipolar spindle 1) kinase (also called Threonine Tyrosine Kinase, TTK), a necessary SAC component (*Weiss and Winey, 1996*; *Abrieu et al., 2001*) that begins high activity shortly before NEBD, (*Kuijt et al., 2020*) by treatment with CFI-402257 (*Liu et al., 2016*; *Mason et al., 2017*). Both control and acentriolar cells treated with CFI-402257 showed reduced mitotic durations (*Figure 1c and c'*) as measured by the time from visible NEBD to visible nuclear envelope reformation. Thus, MPS1 activity is necessary for the extended mitosis observed in acentrosomal cells.

While the majority of control cells treated with CFI-402257 underwent bipolar mitoses, the majority of acentriolar cells treated with CFI-402257 rounded up and spread back out without dividing, producing a single daughter cell (*Figure 1c and d*). Acentriolar cells treated with a different MPS1 inhibitor, MPS1-IN-1 (*Kwiatkowski et al., 2010*), also underwent mitotic events that produced only one daughter cell (*Figure 1—figure supplement 1*). Examination of nuclear morphology revealed that the resultant cells were mononucleate, suggesting that they also failed to undergo nuclear

division; however, the nuclei of acentrosomal cells after undergoing mitosis in CFI-402257 were highly abnormal in morphology (*Figure 1—figure supplement 2*).

To further examine mitosis in acentrosomal cells without MPS1 activity, we generated U2OS cells expressing N-terminally GFP-tagged α-tubulin (*GFP-TUBA1B*) from one allele of the *TUBA1B* locus, made using CRISPR/Cas9-mediated homologous recombination (*Roberts et al., 2017*; *Figure 1—figure supplement 3*). Cells were treated with centrinone for 10 days then were incubated with a low concentration of SiR-Hoechst (50 nM) to label DNA for imaging. In the mitotic cell shown (*Figure 1e*), a single α-tubulin punctum was present alongside condensed chromosomes. While the chromosomes eventually decondensed, a misshapen nucleus formed without cell division occurring.

The phenotypes observed, elongated mitosis and sensitivity of cell division to MPS1 inhibition, were reversible after 10 days of centrinone washout and return to normal centriole number (*Figure 1—figure supplement 4*). The cell division failure phenotype observed in acentriolar CFI-402257-treated cells could be prevented by treatment with proTAME, (*Zeng et al., 2010*) an inhibitor of the APC/C (*Figure 1f*, *Figure 1—figure supplement 1b*). ProTAME treatment resulted in a mitotic arrest of 6 hr or more, regardless of centrosome status, indicating that the rounding up/sitting down behavior of acentrosomal cells treated with MPS1 inhibitor truly represented mitotic entry and exit in the absence of cell division.

## Delayed anaphase onset, rather than specific MPS1 activity, is necessary for successful acentrosomal cell division

We considered whether specifically the kinase activity of MPS1 was required for division in acentrosomal cells or rather its general role in promoting mitotic delay. Aurora B kinase is enriched at kinetochores and chromosomes during mitosis and regulates a variety of SAC and error-correction components (*Gurden et al., 2018*; *Santaguida et al., 2011*; *Roy et al., 2022*; *Liang et al., 2020*). We therefore used inhibition of Aurora B to perturb SAC function independent of MPS1 inhibition. To perturb Aurora B function, we incubated cells with the Aurora B inhibitor AZD1152 (50 nM), also known as Barasertib, (*Mortlock et al., 2007*) and imaged cells live by phase microscopy (*Figure 2a*). Unlike with inhibition of MPS1, both control and acentrosomal cells underwent elongated cell divisions when treated with AZD1152 (*Figure 2a'*). Furthermore, in contrast to treatment with MPS1 inhibitors, both control and acentrosomal cells divided into two daughter cells the majority of the time (*Figure 2a''*). Co-treatment with MPS1 and Aurora B inhibitors resulted in shortened mitosis in the acentrosomal cells and resulted in a similar division failure rate as with MPS1 inhibition alone (*Figure 2a–a''*).

These results suggest that the mitotic delay caused by MPS1 activity is necessary for successful cell division in acentrosomal cells. We tested this by asking whether the phenotype of division failure in acentrosomal cells under MPS1 inhibition could be rescued by elongating mitosis independent of checkpoint activation. APC/C inhibition resulted in increased mitotic arrest (>6 hr) in cells regardless of MPS1 inhibition (*Figure 1e*). However, many of the acentrosomal cells treated with both proTAME and CFI-402257 eventually went on to divide, after the prolonged mitotic arrest. Treatment with both proTAME and CFI-402257 resulted in majority bipolar divisions in acentrosomal cells (*Figure 2b–b''*), in contrast to treatment with CFI-402257 alone (*Figure 1d*). This suggests that the division failure observed in acentrosomal cells treated with MPS1 inhibitor was due to mitotic shortening caused by preemptive anaphase onset and not other activities of MPS1.

## Polyploidy is not sufficient to confer MPS1-inhibition-sensitive division failure

We next asked what defect of an acentrosomal mitosis requires a delay in anaphase onset to complete cell division. Acentrosomal cell divisions are more prone to chromosome missegregation and fragmentation, (*Meitinger et al., 2016*; *Wang et al., 2020*; *Meitinger et al., 2020*) and populations of acentrosomal cells cultured for multiple generations are more likely to be aneuploid (*Wang et al., 2020*). We confirmed that our acentrosomal cells were polyploid by counting chromosome pairs in metaphase spreads (*Figure 3a*). A simple hypothesis would be that cell division in acentrosomal cells is sensitive to MPS1 inhibition due to a requirement for the SAC in cells with increased ploidy. In fact, previous results show that SAC activity was required for the mitotic elongation in tetraploid acentrosomal mouse embryos (*Paim and FitzHarris, 2019*) and that aneuploid cultured cells are more

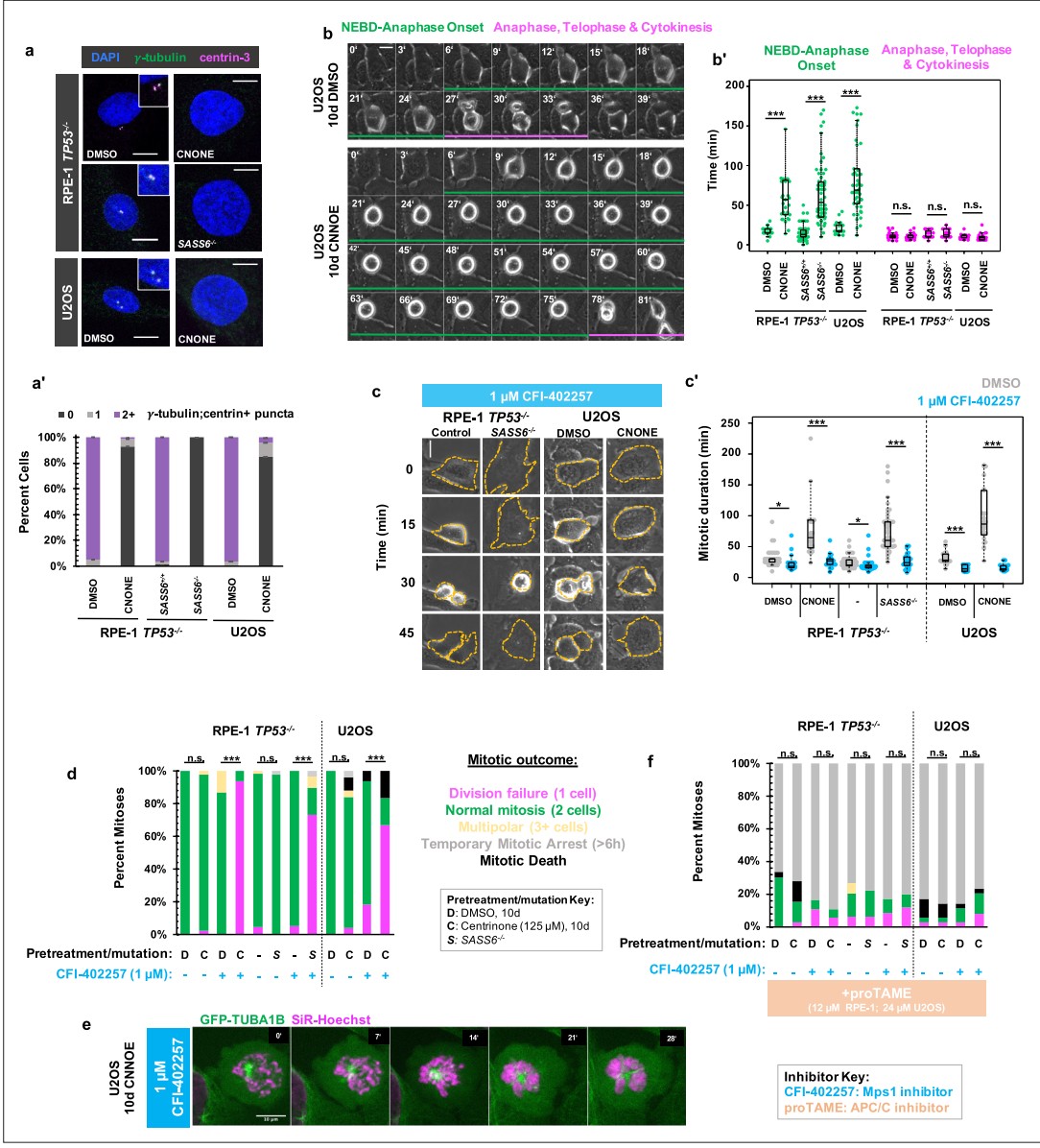

**Figure 1.** MPS1 activity is required for mitotic elongation and cell division in acentrosomal cells. (**a**) Immunofluorescence staining of control and acentriolar RPE-1 *TP53⁻/⁻* or U2OS cells. Acentriolar cells were created by treatment with 125 nM centrinone (CNONE) or by deletion of *SASS6*. DAPI is shown in blue, γ-tubulin in green, and centrin-3 in magenta. Insets are shown for cells in which centrosomes were present. (**a′**) Quantification of (**a**). Graphed are means and S.E.M. Significance was determined by Fisher's exact test. n=100 cells per condition. (**b**) Live phase contrast imaging showing example mitosis in U2OS cells pre-treated with DMSO or centrinone (CNONE) for 10 days. Green bars below cells indicate the duration of nuclear envelope breakdown (NEBD) through the onset of anaphase, while magenta bars below cells indicate the duration of the completion of anaphase, telophase, and cytokinesis. (**b′**) Mitotic durations from NEBD through metaphase (green) and anaphase, telophase, and cytokinesis (magenta) as in (**b**). Time is in minutes. Points represent individual cells; boxplots represent mean and interquartile range. Significance was determined by Welch's *t*-test. *n*≥20 cells per condition. (**c**) Live phase contrast imaging showing example mitosis in U2OS cells pre-treated with DMSO or centrinone (CNONE) for 10 days before being imaging in 1 μM CFI-402257. (**c′**) Quantification of mitotic duration in cells of indicated genotype or treatment in DMSO (grey) or CFI-402257 (blue). Time is in minutes. Points represent individual cells; boxplots represent mean and interquartile range. Significance was determined by Welch's *t*-test. *n*≥25 cells per condition. (**d**) Quantification of daughter cell fate of cells of the given pre-treatment imaged in DMSO or CFI-402257 (1 μM). Shown are the percentages for each fate of the total mitotic observations. Significance was determined by Fisher's exact test. *n*≥40 cells per condition. (**e**) Confocal timelapse imaging of

*Figure 1 continued on next page*

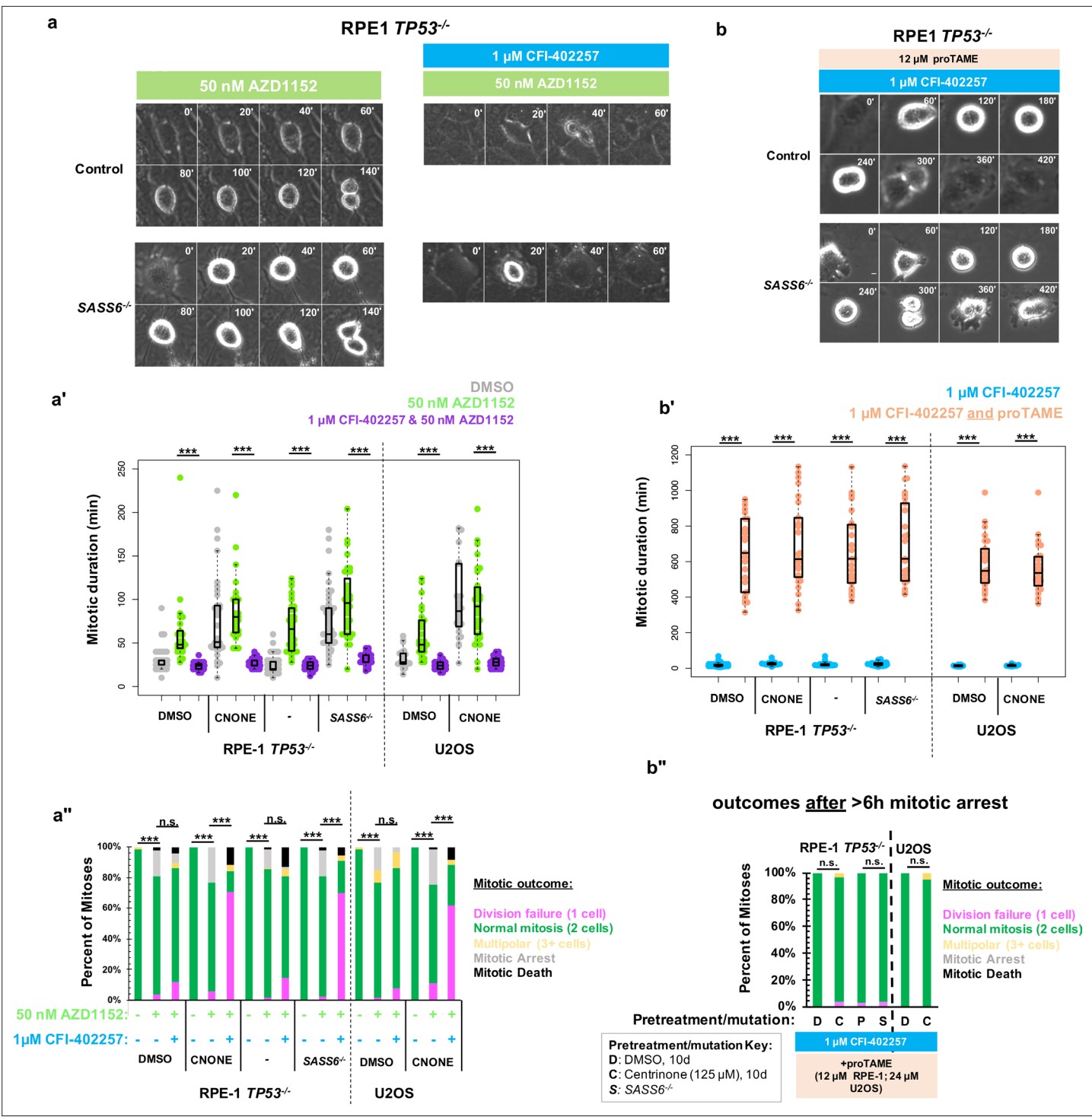

**Figure 2.** Delayed anaphase onset, rather than specific MPS1 activity, is necessary for successful acentrosomal cell division. (**a**) Example mitoses in RPE1 TP53⁻/⁻ or RPE1 *TP53⁻/⁻*; *SASS6⁻/⁻* cells treated with 50 nM AZD1152 with or without 1 µM CFI-402257. (**a'**) Quantification of mitotic duration in cells of indicated genotype or treatment in DMSO (grey), 50 nM AZD1152 (green), or 50 nM AZD1152 and 1 µM CFI-402257 (purple). Points represent individual cells; boxplots represent mean and interquartile range. Significance was determined through Welch's *t*-test. *n*≥47 cells per condition. (**a''**) Quantification of daughter number of cells of the given pre-treatment imaged in DMSO, 50 nM AZD1152 or 50 nM AZD1152 and 1 µM CFI-402257. Shown are the percentages for each fate of the total mitotic observations. *n*≥20 cells per condition. Significance was determined through a Fisher's exact test. (**b**) Example mitoses in RPE1 TP53⁻/⁻ or RPE1 *TP53⁻/⁻*; *SASS6⁻/⁻* cells treated with 1 µM CFI-402257 and proTAME (12 µM for RPE1 cells, 24 µM for U2OS cells). (**b'**) Quantification of mitotic duration in cells of indicated genotype or treatment in 1 µM CFI-402257 (blue) or 1 µM CFI-402257 with 12 µM (RPE1 cells) or 24 µM (U2OS cells) proTAME. Points represent individual cells; boxplots represent mean and interquartile range. Significance was

*Figure 2 continued on next page*

Figure 2 continued

determined through Welch's *t*-test. *n*≥25 cells per condition. (**b''**) Quantification of daughter number of cells of the given pre-treatment imaged in 1 μM CFI-402257 (blue) or 1 μM CFI-402257 with 12 μM (RPE1 cells) or 24 μM (U2OS cells) proTAME after a mitotic arrest of greater than six hours. Shown are the percentages for each fate of the total mitotic observations. *n*≥20 cells per condition. Significance was determined through a Fisher's exact test. In all cases not significant (n.s.) denotes p>0.05 and *** denotes p<0.001. Scale bars: 10 μm.

The online version of this article includes the following source data for figure 2:

**Source data 1.** Source data for *Figure 2a, a", b and b"*.

sensitive to SAC inhibition (*Cohen-Sharir et al., 2021*). We therefore tested the contribution of polyploidy to division failure during MPS1 inhibition. If polyploidy is sufficient for this phenotype in RPE1 and U2OS cells, then polyploid cells with centrosomes should also require a SAC-based delay to divide. We treated RPE1 *TP53⁻ᐟ⁻* cells with 50 μM blebbistatin, a myosin inhibitor (*Straight et al., 2003*), for 12 hr to induce cytokinesis failure, which resulted in a mixed population of binucleate cells resulting from division failure and mononucleate cells that had not entered mitosis during the incubation and served as an internal control for normal ploidy (*Figure 3b*). We then removed the blebbistatin, treated with the MPS1 inhibitor CFI-402257, and assessed division outcomes. Under treatment with CFI-402257, mononucleate RPE1 *TP53⁻ᐟ⁻* cells divided into two daughters the majority of the time

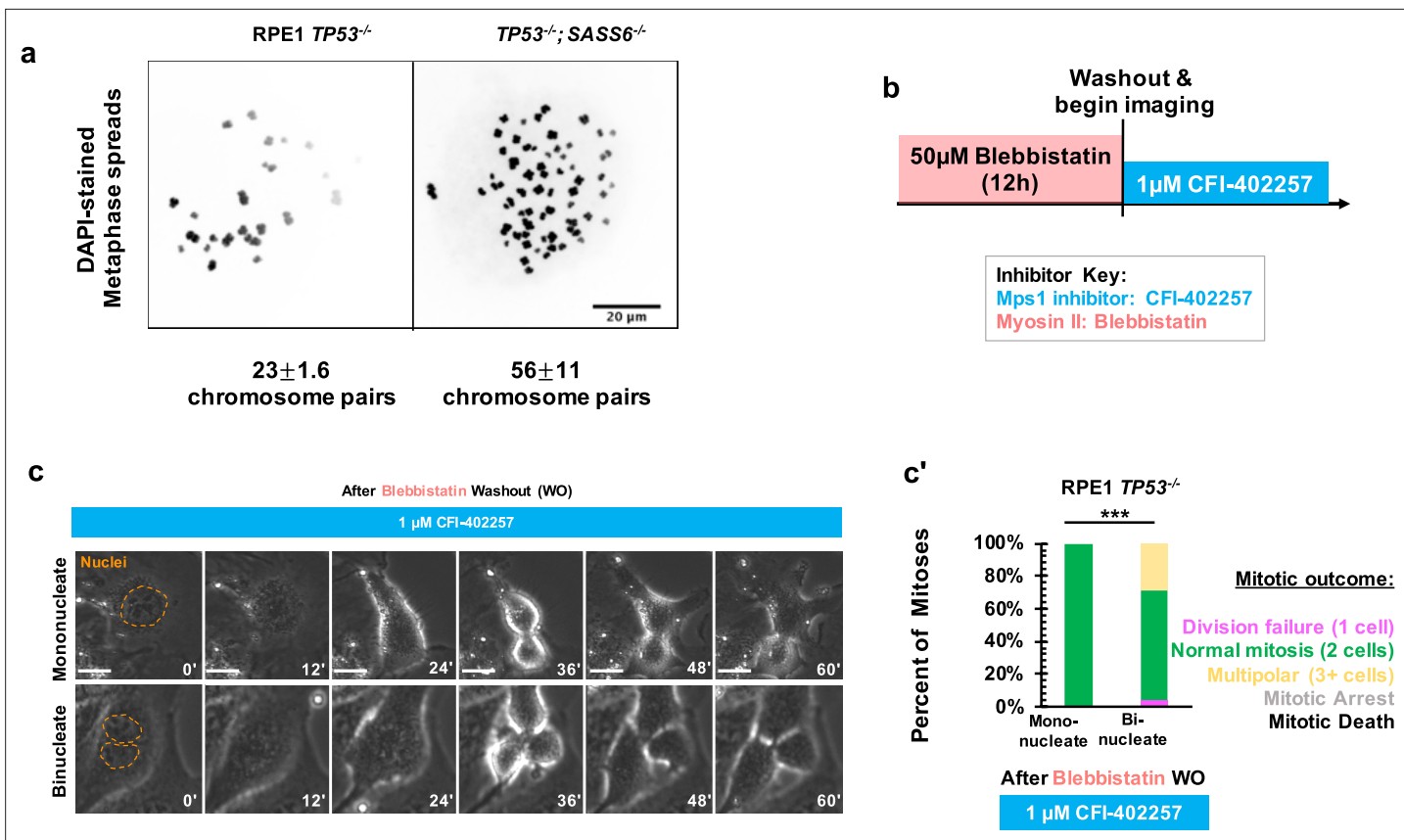

**Figure 3.** Polyploidy is not sufficient to confer MPS1-inhibition-sensitive division failure. (**a**) Metaphase spreads from RPE1 *TP53⁻ᐟ⁻* or RPE1 *TP53⁻ᐟ⁻; SASS6⁻ᐟ⁻* cells. Indicated beneath example images are the mean and standard errors of chromosome pairs for the two genotypes. Chromosomes were stained with DAPI. (**b**) Schematic of experimental setup. REP1 TP53-/- cells were treated with 50 μM blebbistatin for 12 hr, washed, and then treated with 1 μM CFI-402257 and imaged live via phase microscopy. (**c**) Example mitoses in mononucleate or binucleate cells treated with 1 μM CFI-402257 after washout (WO) of blebbistatin. (**c'**) Quantification of mitotic outcome of RPE1 *TP53⁻ᐟ⁻* or U2OS cells imaged in DMSO, 5 μM STLC, 1 μM CFI-402257, or 5 μM STLC and 1 μM CFI-402257 together. Shown are the percentages for each fate of the total mitotic observations. *n*≥48 cells per condition Significance was determined through a Fisher's exact test. In all cases, not significant (n.s.) denotes p>0.05 and *** denotes p<0.001. Scale bars: 10 μm.

The online version of this article includes the following source data for figure 3:

**Source data 1.** Source data for *Figure 3c'*.

*Figure 1 continued*

U2OS cells pretreated for 10 d with 125 nm centrinone and imaged in 1 µM CFI-402257. Shown are endogenously tagged α-tubulin (*GFP-TUBA1B*) and DNA (Sir-Hoechst). (**f**) Quantification of daughter cells fates of cells of the given pre-treatment imaged in DMSO or CFI-402257 (1 µM) with concurrent treatment with proTAME (12 µM for RPE1 cells, 24 µM for U2OS cells). Shown are the percentages for each fate of the total mitotic observations. Significance was determined by Fisher's exact test. $n \geq 40$ cells per condition. In all cases, not significant (n.s.) denotes $p > 0.05$, * denotes $p < 0.05$, ** denotes $p < 0.01$, and *** denotes $p < 0.001$. Scale bars: 10 µm.

The online version of this article includes the following source data and figure supplement(s) for figure 1:

**Source data 1.** Source data for *Figure 1a', b', c', d and f*.

**Figure supplement 1.** Treatment with alternate MPS1 inhibitor MPS1 IN-1 phenocopies treatment with CFI-402257.

**Figure supplement 1—source data 1.** Source data for *Figure 1—figure supplement 1a*.

**Figure supplement 2.** MPS1 inhibition results in abnormal nuclear morphology, but not binuclearity, in acentrosomal cells.

**Figure supplement 2—source data 1.** Source data for *Figure 1—figure supplement 2a*.

**Figure supplement 3.** Creation and validation of *GFP-TUBA1B* U2OS cells.

**Figure supplement 3—source data 1.** Source data for *Figure 1—figure supplement 3a–d*.

**Figure supplement 3—source data 2.** Source data for *Figure 1—figure supplement 3a–d* with raw blots.

**Figure supplement 4.** Division failure in acentrosomal cells due to Mps1 inhibition is reversible after centrosome return.

**Figure supplement 4—source data 1.** Source data for *Figure 1—figure supplement 4a', b, and c*.

(*Figure 3c and c'*). Binucleate RPE1 *TP53⁻/⁻* cells divided the majority of the time, into either two or multiple (3+) daughters (*Figure 3c and c'*). Thus, polyploidy alone, as generated by cytokinesis failure, was not sufficient to cause division failure when MPS1 is inhibited.

## Early mitotic MTOC separation is necessary for cell division in the absence of SAC-mediated mitotic delay

We next tested whether the division failure phenotype resulted from temporal differences in establishing bipolarity in the absence of centrosomes. This would be consistent with the monopolar spindle seen under MPS1 inhibition in *Figure 1e*, and with previous work showing delays in bipolar organization of microtubule-minus-end-associated proteins during early mitosis in acentrosomal cells (*Watanabe et al., 2020*; *Chinen et al., 2021*; *Chinen et al., 2020*). We used acentrosomal and control U2OS *GFP-TUBA1B* cells to directly visualize microtubule organization in early mitosis. While control cells had two tubulin-dense MTOCs visible before NEBD, acentrosomal cells only established two MTOCs well after NEBD and chromosome condensation (*Figure 4a and a'*).

One possible explanation for the robustness of cell division in cells with centrosomes is that having two separate and functional MTOCs at the beginning of mitosis facilitates timely division, without requiring a SAC-mediated delay. We therefore tested whether a monopolar spindle, such as in acentrosomal cells early in mitosis, is sufficient to cause division failure upon MPS1 inhibition, even in the presence of centrosomes. We treated RPE1 or U2OS cells containing centrosomes with (+)-S-Trityl-L-cysteine (STLC), an inhibitor of the bipolar kinesin Eg5 (also known as Kif11 or kinesin-5), resulting in monopolar mitotic spindles (*Skoufias et al., 2006*; *Ogo et al., 2007*). As expected, treatment with STLC alone caused mitotic arrest (*Figure 4b and b'*); however, inhibiting MPS1 in addition to Eg5 resulted in exit from mitotic arrest and division failure (*Figure 4b and b'*) similar to the phenotype observed in acentrosomal cells.

To better understand how centrosome separation in early mitosis allows normal mitosis to occur without MPS1-dependent delay, we imaged Eg5-inhibited cells, with or without MPS1 inhibition, by confocal microscopy. Cells treated with both Eg5 and MPS1 inhibitors formed monopolar spindles but then exited from mitosis (*Figure 4c*), failing to divide. However, frequently some chromosomes became extruded (*Figure 4c*, asterisk) during what appeared to be a highly asymmetric division, with both centrosomes remaining unseparated in the primary cell. In addition, a midbody-like structure (*Figure 4c*, arrow) formed between the main resultant cell and the extruded portion. Taken together,

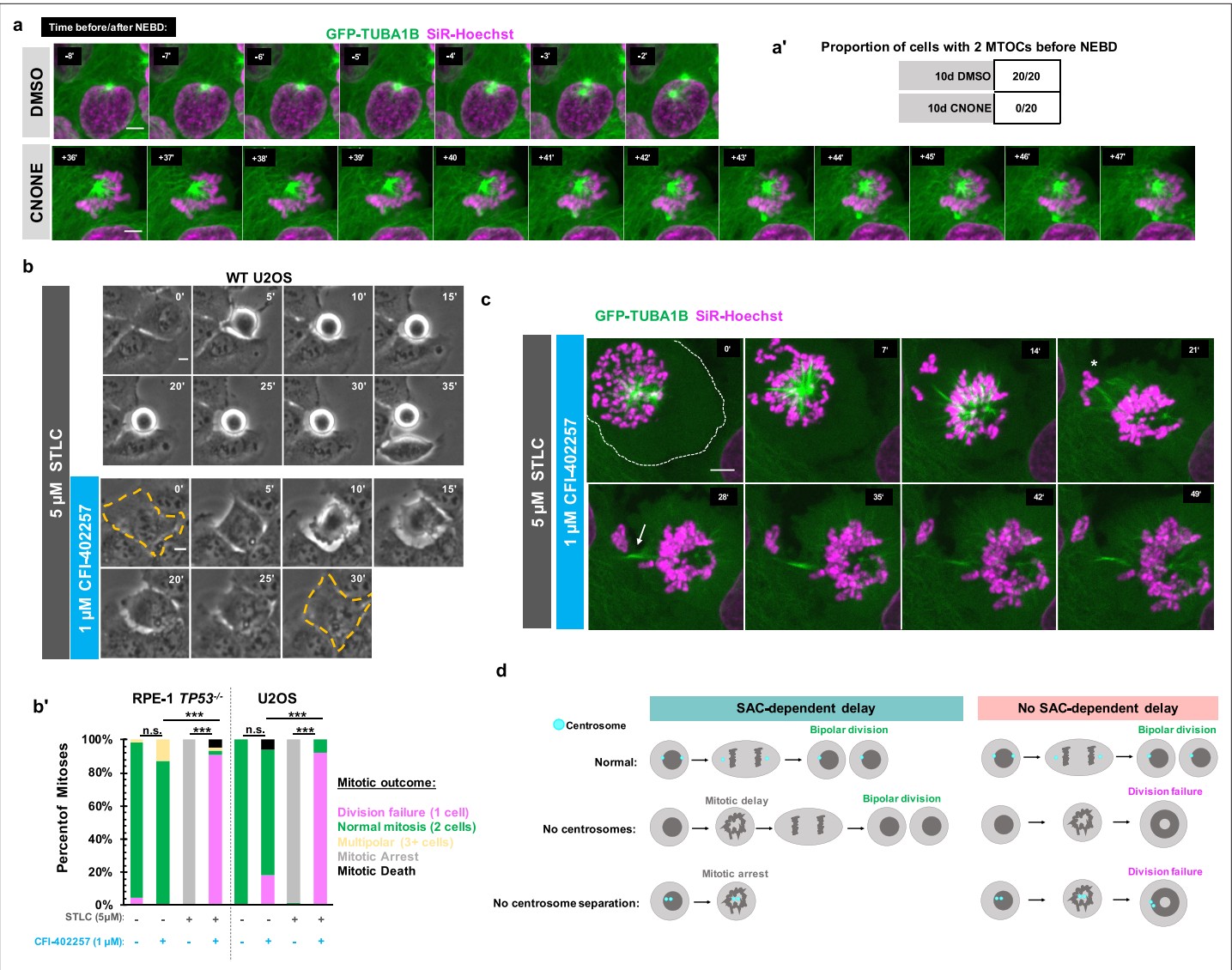

**Figure 4.** Early mitotic MTOC separation is necessary for cell division in the absence of SAC-mediated mitotic delay. (**a**) Confocal timelapse imaging of U2OS cells pretreated for 10 days with either DMSO or centrinone (125 nM). Shown are endogenously tagged α-tubulin (*GFP-TUBA1B*) and DNA (Sir-Hoechst). Time indicates minutes before (-) or after (+) NEBD. (**a'**) Proportion of cells with 2 foci of microtubules before NEBD based on experiments as in (**a**). (**b**) Live phase imaging of RPE1 *TP53*-/- or U2OS cells imaged in 1 µM CFI-402257 or 5 µM STLC and 1 µM CFI-402257 together. (**b'**) Quantification of mitotic outcome of RPE1 *TP53*-/- or U2OS cells imaged in DMSO, 1 µM CFI-402257, 5 µM STLC, or 5 µM STLC and 1 µM CFI-402257 together. Shown are the percentages for each fate of the total mitotic observations. Significance was determined through a Fisher's exact test. *n*≥50 cells per condition (**c**) Confocal timelapse imaging of U2OS cells imaged in 5 µM STLC with 1 µM CFI-402257. Shown are endogenously tagged α-tubulin (*GFP-TUBA1B*) and DNA (Sir-Hoechst). Asterisk indicated extruded chromosomes. Arrow indicated midbody-like α-tubulin structure. (**d**) Graphical summary of results. In all cases, not significant (n.s.) denotes p>0.05 and *** denotes p<0.001. Scale bars: 10 µm.

The online version of this article includes the following source data for figure 4:

**Source data 1.** Source data for *Figure 4b'*.

these results show that the early separation of centrosomes prevents monopolar spindles from persisting, allowing a timely completion of mitosis without a requirement for SAC-mediated delay.

Here, we show that acentrosomal cells rely on SAC-mediated delay, not simply to correct chromosome attachment errors, but to build a bipolar spindle and successfully divide into two daughters (*Figure 4d*). Because centrosomes form two MTOCs very early in mitosis and can be efficiently separated into two spindle poles by Eg5 activity, they provide a 'timely two-ness' to mitotic organization. This rapid establishment of bipolarity normally allows cells to successfully divide in the absence of

a SAC-induced delay. The finding of such 'cross-talk' between centrosomes and the SAC leads to several questions about spindle organization and function. For instance, it it is unclear why acentrosomal cells, whether created in culture or during female meiosis in many metazoans, are prone to chromosome missegregation despite the ability to assemble bipolar spindles (*Meitinger et al., 2016*; *Wang et al., 2020*; *Meitinger et al., 2020*; *Holubcová et al., 2015*; *Yoshida et al., 2020*). It is known that SAC activity weakens over time during prolonged arrest, characterized by decreasing levels of cyclin B globally and MAD2 at kinetochores, thus enabling mitotic slippage (*Lok et al., 2020*; *Brito and Rieder, 2006*; *Brito et al., 2008*). Therefore, it will of interest to determine whether the efficiency that centrosomes provide to spindle bipolarity enables important temporal coordination between SAC-mediated delay and correction of spindle defects by ensuring that these corrections occur during a time window when global cyclin B and kinetochore MAD2 levels are still elevated. On a more translational note, MPS1 inhibitors are under investigation as chemotherapeutic agents (*Maia et al., 2018*; *Simon Serrano et al., 2020*; *Atrafi et al., 2021*; *Martinez et al., 2015*), and the presence of acentrosomal cells in cancers is now being appreciated (*Wang et al., 2020*; *Morretton et al., 2022*) Given our results, centrosome presence or loss may be a relevant clinical marker to consider when predicting effects of MPS1 inhibition on division outcome.

## Materials and methods

### Cell lines and culture

hTERT RPE-1; *TP53*[-/-] and hTERT RPE-1; *TP53*[-/-]; *SASS6*[-/-] were a gift from Meng-Fu Bryan Tsou (Sloan Kettering Institute) and described previously (*Wang et al., 2015*; *Tanos et al., 2013*). U2OS cells were acquired from the ATCC. hTERT RPE-1 cells of all genotypes were cultured in DMEM/F-12 (with L-glutamine and 15 mM HEPES, Corning) supplemented with 10% fetal bovine serum (FBS; Gemini Bio-Products), and 1% Pen-Strep (final concentrations: 100 U/mL penicillin G and 0.1 mg/mL streptomycin, Gemini Bio-Products). U2OS cells were cultured in DMEM (with L-glutamine, 4.5 g/L glucose and sodium pyruvate, Corning) supplemented with 10% CCS and 1% Pen-Strep. All cells were cultured at 37 °C with 5% CO2. Cells were routinely tested for mycoplasma with MycoAlert Mycoplasma Detection Kit (Lonza) as per manufacturer instructions.

### Drugs and live stains

Drugs were used the indicated concentrations in figures and figure legends and came from the following sources: CFI-402257 (Caymen Chemical Company, stock concentration: 4 mM in DMSO), proTAME (R&D Systems, stock concentration: 20 mM in DMSO), S-Trityl-L-cysteine (STLC) (Sigma-Aldrich, stock concentration: 10 mM in DMSO); MPS1 IN-1 (Caymen Chemical Company, stock concentration: 10 mM in DMSO); AZD1152-HQPA (also called Barasertib-HQPA, Caymen Chemical Company, stock concentration: 50 mM in DMSO); (S)-(-)-Blebbistatin (Toronto Research Chemicals, stock concentration 100 mM in DMSO). Centrinone, a generous gift from Karen Oegema (UC San Diego), was always used at 125 nM (stock concentration: 125 µM in DMSO). For live imaging, SiR-DNA/SiR-Hoechst (Spirochrome) was used at 50 nM (stock concentration: 5 mM in DMSO).

### Live phase imaging

Cells were plated $5 \times 10^4$ cells/well in glass bottom 12-well plates (No. 1.5 Coverslip, 14 mm Glass Diameter, Uncoated 12-well plate, Mattek) or 24-well plated (No. 1.5 Coverslip, 14 mm Glass Diameter, Uncoated 24-well plate, Mattek) to facilitate simultaneous imaging of various drug concentrations. Phase images were acquired with a Keyence microscope with either a Nikon Plan Fluor 20×/0.45 NA Ph1 objective or a Nikon Plan Fluor 10x/0.3 NA Ph1 objective. Temperature was maintained at 37 °C in a humidified chamber with 5% $CO_2$ flowed in. Images were acquired every 4 or 5 min depending on number of positions acquired.

### Confocal microscopy

For both fixed and live cells, samples were imaged with an Zeiss Axio Observer microscope (Carl Zeiss) with a confocal spinning-disk head (Yokogawa Electric Corporation, Tokyo, Japan), PlanApoChromat 63×/1.4 NA objective, and a Cascade II:512 electron-multiplying (EM) CCD camera (Photometrics,

Tucson, AZ) or PRIME: BSI backside illuminated CMOS camera run with SlideBook 6 software (3i, Denver, CO). Excitation lasers were 405 nm, 488 nm, 661 nm, and 640 nm.

## CRISPR endogenous tagging of *TUBA1B* with mEGFP in U2OS cells

To create U2OS cells expressing mEGFP-TUBA1B, we followed the same CRISPR/Cas9 gene editing design as in *Roberts et al., 2017* Briefly, U2OS cells were co-transfected with pLentiCRISPRv2 (*Sanjana et al., 2014*; Addgene plasmid # 52961; RRID:Addgene_52961) expressing a gRNA targeted against *hTUBA1B* (gGATGCACTCACGCTGCGGGA) and an mEGFP plasmid with 1 kb homology arms (AICSDP-4:TUBA1B-mEGFP, Allen Institute, Addgene plasmid # 87421; RRID:Addgene_87421) (*Roberts et al., 2017*) mutated for the gRNA binding site. Two days after transfection, cells were split into a fresh plate with 1 µg/mL puromycin. After 2 days of selection, cells were placed into fresh media. Single-cell clones were obtained by serial dilution, and clones were screened by PCR with the following primers:

> forward primer 5'-GGGGTGCTGGGTAAATGGAG
> reverse primer 5'- CGGTTTAGGATGGGAAGGTAACATT.

Clones were then assessed via live imaging for GFP fluorescence, western blot for GFP tagging of α-tubulin, the ability of this tagged α-tubulin to be acetylated, and the absence of Cas9 protein in the established clone, and immunofluorescence for localization of GFP-TUBA1B to spindle microtubules. The chosen clone was sequenced and found to have one allele edited with the expected insertion and one allele unedited at the *TUBA1B* locus.

## Live fluorescent imaging of microtubules and DNA

U2OS *GFP-TUBA1B* cells were plated into 35 mm glass bottom dishes (Fluorodish) in normal grown medium. Thirty min before onset of imaging, medium was changed to phenol free DMEM (with L-glutamine, 4.5 g/L glucose and sodium pyruvate, Corning) containing 10% CCS (Gemini bio-Products), 50 nM SiR-Hoechst (Spirochrome), and any drug treatments (STLC or CFI-402257). Temperature was maintained at 37 °C, and the imaging dish was kept in a chamber into which humidified 5% $CO_2$ was flowed in. Images were acquired every minute by confocal microscopy (488 nm and 640 nm laser lines).

## Antibodies and fixed stains

Primary antibodies used for immunofluorescence (IF) and western blot (WB): mouse IgG1 anti-$\alpha$-tubulin, clone DM1A (1:1000IF, 1:5000 WB, Sigma-Aldrich T6199; RRID:AB_477583); mouse IgG2$_b$ anti-centrin3, clone 3e6 (1:1000 IF, Novus Biological H00001070-M01, RRID:AB_537701); mouse IgG1anti-$\gamma$ -tubulin, clone GTU-88 (1:1000 IF, Sigma-Aldrich T5326; RRID:AB_477584); rabbit anti-CP110 (1:200 IF, Proteintech 12780–1-AP; RRID:AB_10638480); mouse IgG2$_b$ anti-Lys-40-acetylated $\alpha$-tubulin, clone 6-11B-1 (1:1000 IF, 1:10,000 WB, Sigma-Aldrich T6793; RRID:AB_477585); mouse IgG2$_a$ anti-GFP, clone 3e6 (1:1000 IF, 1:5000 WB, Thermo Fischer A-11120; RRID:AB_221568); mouse IgG1 anti-Cas9, clone 7A9 (1:1000 WB, BioLegend 698301, RRID:AB_2715788). Secondary antibodies conjugated to AlexaFluor (Thermo Fisher/Invitrogen) were used for IF. Secondary antibodies conjugated to IR dyes (LiCOR) were used for western blots. REVERT total protein stain (LiCOR) was used as indicated by the manufacturer.

## Immunofluorescence

Before use, #1.5 coverslips were washed with 70% ethanol and dried under UV light. Cells were fixed for 7 min in 100% 20 °C methanol. Cells were washed 3 x at room temperature (RT) with PBS and blocked for 10 min with PBS-BT (PBS +3% BSA+0.1% Triton-X). Primary antibodies diluted in blocking solution were incubated for 1 hr at RT. Cells were then washed 3x10 min in blocking solution and then incubated with secondary antibodies, also in blocking solution, for 45 min at RT. Cells were washed 2x10 min with blocking solution, and 1x10 min with PBS (containing 5 µg/ml DAPI (4',6-diamidino-2 -phenylindole), if used). Samples were then mounted in Mowiol mounting medium (Polysciences) in glycerol containing 2.5% 1,4-diazabicyclo-(2,2,2)-octane (DABCO, Sigma-Aldrich) antifade.

## Metaphase spreads

Cells were incubated for 3 hr in 10 ng/µL nocodazole in normal growth medium. Cells were then trypsinized and pelleted and resuspended in ice cold 0.56% KCl solution, incubated at RT for 7 min for

hypotonic lysis to occur, then pelleted. Pellets were then disrupted and cells were resuspended and fixed in fixative (75% v/v methanol; 25% v/v glacial acetic acid) added dropwise. Cells were pelleted and resuspended in fixative. 20 µL drops were placed onto ethanol-cleaned slides, which were left to dry. Spreads were then overlayed by Mowiol mounting medium (Polysciences) in glycerol containing 2.5% 1,4-diazabicyclo-(2,2,2)-octane (DABCO, Sigma-Aldrich) antifade containing 5 µg/ml DAPI (4',6-diamidino-2-phenylindole). #1.5 coverslips were placed on top, and samples were imaged via confocal microscopy.

## SDS-PAGE and western blot

Samples were separated by 12% SDS-PAGE reducing gels (5% stacking gels) and transferred to nitro-cellulose membrane at 4 °C. Blots were blocked in 5% (w/v) non-fat dry milk in TBST (TBS +1% Tween-20). Primary and secondary antibody staining were performed in blocking solution. When used, total protein on nitrocellulose membranes was stained via REVERT Total Protein (LiCOR) as per manufacturer instructions and imaged before immunodetection. REVERT-stained blots and immunoblots were imaged using a LiCOR Odyssey imager.

## Statistical analysis

For categorical data, significance was determined a Fisher's exact test. For quantitative data, significance was determined via Welch's *t*-test. The test performed is also specified in the figure legend. In all cases, not significant (n.s.) denotes $p > 0.05$; * denotes $p < 0.05$; ** denotes $p < 0.01$; *** denotes $p < 0.001$. Statistical tests were performed in R Studio. SEM was calculated in Excel or R-Studio as one standard deviation divided by the square-root of the *n*. The given *n* in figure legends refers to biological replicates from two or more technical replicates. Sample size was determined using the maximum number of cells practically observable given constraints of the experiments. For exact *n*, please consult the source data spreadsheet.

## Novel materials availability statement

The newly created U2OS *mEGFP-TUBA1B* cell line generated in this work is available through contacting Tim Stearns (Rockefeller University) and/or Jennifer T. Wang (Department of Biology, Washington University in St. Louis).

## Acknowledgements

Centrinone was a gift of Karen Oegema (University of California, San Diego). hTERT-RPE1 TP53[-/-] and TP53[-/-];SASS6[-/-] cells were a gift from Meng-Fu Bryan Tsou (Sloan Kettering Institute). The plasmid AICSDP-4:TUBA1B-mEGFP was a gift from Allen Institute for Cell Science (Addgene plasmid # 87421; RRID:Addgene_87421). The plasmid plentiCRISPRv2 was a gift from Feng Zhang (Addgene plasmid # 52961; RRID:Addgene_52961). We thank Alex Long (Stanford University), Ljiljana Milenkovic (Stanford University), and Sabrina Ergun (Princeton University) for critical feedback on the manuscript. KF was supported by the National Institutes of Health (NIH) under award number T32GM007276. JTW was supported by the National Institute of General Medical Sciences of the NIH under award number K99GM131024. This work was supported by the NIH under grant R35GM130286 to TS. The content is solely the responsibility of the authors and does not necessarily represent the official views of the NIH.

## Additional information

### Funding

| Funder | Grant reference number | Author |
| --- | --- | --- |
| National Institutes of Health | R35GM130286 | Tim Stearns |
| National Institute of General Medical Sciences | K99GM131024 | Jennifer T Wang |

| Funder | Grant reference number | Author |
| --- | --- | --- |
| National Institutes of Health | T32GM007276 | KC Farrell |

The funders had no role in study design, data collection and interpretation, or the decision to submit the work for publication.

## Author contributions

KC Farrell, Jennifer T Wang, Conceptualization, Data curation, Formal analysis, Investigation, Methodology, Writing – original draft, Writing – review and editing; Tim Stearns, Conceptualization, Supervision, Funding acquisition, Project administration, Writing – review and editing

## Author ORCIDs

KC Farrell (ID) http://orcid.org/0000-0002-1767-3629
Jennifer T Wang (ID) https://orcid.org/0000-0002-8506-5182
Tim Stearns (ID) https://orcid.org/0000-0002-0671-6582

Reviewer #1 (Public Review): https://doi.org/10.7554/eLife.84875.3.sa1
Reviewer #2 (Public Review): https://doi.org/10.7554/eLife.84875.3.sa2
Author response https://doi.org/10.7554/eLife.84875.3.sa3

# Additional files

## Supplementary files

• MDAR checklist

## Data availability

All data generated or analyzed in this work are included in the manuscript and the supporting source data files.

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
