## [Editor Report · eLife assessment]

This work explores how centrosomes, which function as the primary microtubule organizing center in animal cells, regulate cell division by examining the process in cells in which centrosome formation has been inhibited. The carefully conducted experiments provide **convincing** support for the **important** observation that elongated, but successful, mitosis observed in cells lacking centrosomes is due to delays in cell cycle progression.

---

## [Referee Report · Reviewer #1 (Public Review)]

In their manuscript "Spindle assembly checkpoint-dependent mitotic delay is required for cell division in absence of centrosomes," Farrell and colleagues employ carefully chosen approaches to assay mitotic timeliness in the absence of centrosomes in mammalian culture cells, namely the mechanism behind it and its function. The authors acknowledge prior work well and present their data succinctly, clearly, and with a clear logical flow of experiments. The experiments are thoughtfully designed and presented, with appropriate controls in place.

The authors' model whereby centrosome separation and its early definition of poles mediates timely mitosis without relying on a SAC-dependent delay is compelling, and the authors' data are consistent with it. They show using two different MPS1 inhibitors that acentrosomal cell division fails, supporting their claims that a SAC-dependent delay is required in these instances. Furthermore, they show that reintroducing a time delay is sufficient to restore cell division, but inhibiting a different aspect of SAC function does not restore cell division. Next, the authors rule out polyploidy as a potential confounding factor for requiring a SAC-dependent delay, and instead demonstrate that inhibiting centrosome separation by Eg5 inhibition is sufficient to require this delay for mitotic progression. The authors' findings overall support their proposed model.

Probing what aspects of centrosomes protect against a requirement for SAC-dependent delays would strengthen the work and specifically the conclusion that centrosomes provide "two-ness". For example, the authors could examine division in a population of cells with only one centrosome. Seeing some restoration of mitotic progression in the absence of SAC-dependent delays would suggest that even one centrosome with uninhibited Eg5 is sufficient to negate SAC-dependent delays, and would limit models for what exactly centrosomes contribute. This would help disentangle the roles of actual centrosomes and their biochemical cues, Eg5-driven centrosome separation, and early definition of poles on mitotic progression in the absence of SAC-dependent delays. Making a high fraction of cells with one centrosome could be achieved by using centrinone for a shorter time.

Comments on revised version:

The main point from the initial review does not appear to be addressed in the revised version. As such, the comments on the revised version remain the same.

---

## [Referee Report · Reviewer #2 (Public Review)]

Centrosomes are an integral part of cell division in most animal cells. There are notable exceptions, however, such as oocytes and plants. In addition, some animal cells can be engineered to lack centrosomes yet they can still manage to complete mitosis. This paper uses a couple of methods (PLK4 inhibition and deletion of SASS6) to remove centrosomes from an immortalized cell line. Indeed, a strength of the paper is that similar results are obtained using both protocols to generate acentrosomal cells. The authors find acentrosomal cells take longer to divide, mostly due to a longer metaphase. The paper is based on the finding that inhibition of MPS1 results in a failure to divide, and the hypothesis that a SAC - dependent delay is required for these acentrosomal cells to divide.

The finding that MPS1 inhibition results in a failure to division is interesting. This is investigated by analyzing cells where AurB, APC or Eg5 (to generate monastral spindles) have been inhibited. My concerns are that the results are not conclusive that the effect of MPS1 is on cell cycle regulation. There is not enough data to make a conclusion as to why inhibition of MPS1 results in cell division failure.

1. An example is how to interpret the effect of Aurora B inhibition, which does not block acentrosomal cell division. If Aurora B is required for SAC activity, it suggests this effect of MPS1 may be a function other than SAC. Given the complexity of the SAC, it would be informative to test other SAC components. Instead, the authors conclude that the mitotic delay caused by MPS is required for acentrosomal cell division. I don't think they have ruled out, or even addressed other functions of MPS1.

2. The authors find that when both the APC and MPS1 are inhibited, the cells eventually divide. These results are intriguing, but hard to interpret. The authors suggest that the failure to divide in MPS1-inhibited cells is because they enter anaphase, and then must back out. This is hard to understand and there is not data supporting some kind of aborted anaphase. Is the division observed with double inhibition some sort of bypass of the block caused by MPS1 inhibition alone? It is not clear why inhibition of APC causes increased cell division when MPS1 is inhibited.

3. The authors characterize MTOC formation in these cells, which is also interesting. MTOCs are established after NEB in acentrosomal cells. Indeed, forming these MTOCs is probably a key mechanism for how these cells complete a division, like mouse oocytes.

Following this, the results with inhibiting Eg5 are interesting. The double inhibition of MPS1 and Eg5 results in division failure, like MPS1 inhibition in acentrosomal cells. Thus, there is a cell division block when the centrioles fail to divide. This result raises the possibility that failure to make a bipolar spindle, or the presence of a monopolar spindle, is the problem. In the absence of a bipolar spindle, a SAC induced delay is required for spindle assembly. This is a possibility but there are multiple interpretations of these results. Primarily, these results do not show the MPS1 effect on acentrosomal cells is SAC related. That a SAC mediated delay is required for acentrosmomal spindle assembly is not the only conclusion.

Comments on revised version:

It appears that very few changes have been made. These are all suggestions in the writing and interpretation.

It's disappointing the most of the readouts are cell division success. There is a lack of data about what happens in the MPS1 knockdowns, such as microtubule attachment to KTs and localization/ activity of checkpoint proteins or downstream factors. More mechanistic insights may be found by testing other checkpoint proteins or assaying more metrics for spindle assembly and cell cycle progression. Or if inducing cell cycle delay suppresses the MPS1 effect. These experiments would implicate cell cycle factors as being required for acentrosomal spindle assembly while ruling out a requirement for MPS1 in spindle assembly.

The paper is well written. But some of the terminology could be improved and some descriptions of the cytology are confusing. Some clear definitions of terms may help. The authors refer to an "extended mitosis" (line 73) and then "exit in the absence of cell division" (line 96) when MPS1 is inhibited. Both are misleading and don't tell the full story. These cells attempt to divide and then fail, resulting in one cell. Use of terms like "spread back out", "rounding up", and "sitting down" seems like jargon and should at least be defined. The term "timely two-ness" (line 23-24) is also not helpful. A brief discussion of data on MPS1 function in mouse and fly acentrosomal meiosis might be appropriate. A comparison would be interesting since loss of MPS1 in acentrosomal oocytes does not have such a drastic block in cell division.

---

## [Author Response]

The following is the authors’ response to the original reviews.

We thank the reviewers for a careful review of the manuscript and for their comments, which we address below.

**Reviewer #1:**
(1) …the authors could examine division in a population of cells with only one centrosome. Seeing some restoration of mitotic progression in the absence of SAC-dependent delays would suggest that even one centrosome with uninhibited Eg5 is sufficient to negate SAC-dependent delays, and would limit models for what exactly centrosomes contribute.

We agree that the one-centrosome question (i.e. whether cells with a single centriole, and therefore a single centrosome, have the same SAC dependence) would be interesting to address. It is known that cells with a single centriole generated through centrinone treatment also have elongated mitoses, like cells lacking centrioles (see Chinen, et. al. 2021, compare Fig 2C to Fig 2D), We have tried this experiment in RPE-1 cells with preliminary results confirming that there is a mitotic delay. It is not known whether this delay requires SAC activity, and we hope to address that in future work. In addition, we note that we show in Fig. 4b-c that cells with the normal centrosome number but with a single focus of microtubules due to Eg5 inhibition, were also sensitive to MPS1 inhibition. This suggests that centrosome presence alone cannot overcome the requirement for SAC activity, rather, the centrosomes need to be able to separate in a timely fashion.

**Reviewer #2:**
(1) An example is how to interpret the effect of Aurora B inhibition, which does not block acentrosomal cell division. If Aurora B is required for SAC activity, it suggests this effect of MPS1 may be a function other than SAC. Given the complexity of the SAC, it would be informative to test other SAC components. Instead, the authors conclude that the mitotic delay caused by MPS is required for acentrosomal cell division. I don't think they have ruled out, or even addressed other functions of MPS1.

We agree that it is possible that functions of the MPS1 kinase other than those involved in the SAC could be important. Although we have not directly tested other SAC components, we did “mimic” SAC activity by delaying anaphase onset using APC/C inhibition while also inhibiting MPS1 (Fig. 2b-b’’). The fact that this restored division suggests that it is the SAC function of MPS1 kinase activity that is relevant to this delay.

(2) The authors find that when both the APC and MPS1 are inhibited, the cells eventually divide. These results are intriguing, but hard to interpret. The authors suggest that the failure to divide in MPS1-inhibited cells is because they enter anaphase, and then must back out. This is hard to understand and there is not data supporting some kind of aborted anaphase. Is the division observed with double inhibition some sort of bypass of the block caused by MPS1 inhibition alone? It is not clear why inhibition of APC causes increased cell division when MPS1 is inhibited.

As described in the response to (1), we believe that reinstating the delay to anaphase onset by APC/C inhibition provided the time needed to establish a functional bipolar spindle even in the absence of the SAC, and that cells eventually overcome the proTAME block and proceed through mitosis, as observed in control cells in our experiments. We note that we chose concentrations of proTAME specifically for each cell line (RPE-1 and U2OS) that would result only in a temporary block, following on the work of Lara-Gonzalez and Taylor (2012), who reported similar findings for HeLa cells.

(3) The authors characterize MTOC formation in these cells, which is also interesting. MTOCs are established after NEB in acentrosomal cells. Indeed, forming these MTOCs is probably a key mechanism for how these cells complete a division, like mouse oocytes.

We agree that the observed intermediates of MTOCs are interesting and likely crucial to the mechanism of cell division in acentrosomal somatic cells. We are investigating further the differences and similarities between somatic cell MTOC formation in the absence of centrosomes and the naturally-occurring form of that process in oocytes.